# META-LORA: DATA-EFFICIENT MULTI-TASK FINE-TUNING FOR LARGE LANGUAGE MODELS

## ABSTRACT

Low-Rank Adaptation (LoRA) has emerged as one of the most widely used parameter-efficient fine-tuning (PEFT) methods for adapting large language models (LLMs) to downstream tasks. While highly effective in single-task settings, it struggles to efficiently leverage inter-task knowledge in complex multi-task learning scenarios, often requiring substantial task-specific data to achieve optimal performance. To address this limitation, we introduce META-LORA, a two-stage optimization framework that significantly improves data efficiency in multi-task adaptation. In the first stage, task-specific LoRA adapters are learned using only a few samples from each involved dataset, enabling rapid adaptation without large-scale supervision. In the second stage, the shared LoRA adapter is updated by aggregating gradients from multiple tasks to promote knowledge transfer across tasks, further reducing data usage by leveraging common patterns. In both multi-task learning and multilingual learning scenarios, our method matches or surpasses the performance of traditional full-data LoRA fine-tuning approaches, while using significantly less task-specific data.

## 1 INTRODUCTION

Large language models (LLMs) have transformed natural language processing by achieving state-of-the-art results on tasks from text generation to complex reasoning (Brown et al., 2020; Devlin et al., 2019; Chang et al., 2024b). However, the sheer scale of these models, which often encompass billions of parameters, renders full-parameter fine-tuning prohibitively expensive in both computational and memory requirements, especially when adapting to multiple tasks simultaneously (Han et al., 2024). As real-world applications increasingly demand multi-task capabilities, methods that reduce resource overhead while preserving performance have become critical. Parameter-efficient fine-tuning (PEFT) techniques (Hu et al., 2022; Rebuffi et al., 2017a; Li & Liang, 2021; Chang et al., 2024a), which add only lightweight adaptation modules to a frozen base model, offer a promising solution by slashing trainable parameters from hundreds of millions to mere thousands dramatically cutting GPU memory footprint and speeding up training without sacrificing modularity across tasks.

While PEFT approaches like LoRA (Hu et al., 2022) and its multi-task extensions (e.g., R-LoRA (Liu et al., 2025b), HydraLoRA (Tian et al., 2024)) deliver substantial efficiency gains in single-task scenarios, they still rely on large volumes of labeled data when scaled to many tasks resulting in poor data efficiency. For example, HydraLoRA reports that fine-tuning on just 43 tasks required over 320,000 in-domain examples to achieve satisfactory performance on downstream tasks (Tian et al., 2024). This heavy data demand undermines the very efficiency gains PEFT seeks to provide in multi-task settings and highlights the pressing need for methods that balance both parameter- and data-efficiency in large-scale multi-task adaptation.

Data-efficient techniques like coreset selection (Liu et al., 2023; Xia et al., 2024) or data pruning (Azeemi et al., 2023) excel at trimming down data for a single task by homing in on the "most informative" examples, but in a multi-task LLM scenario, this narrow focus comes at the expense of broader generalization across many objectives. By optimizing for task-specific highlights, these methods tend to under-represent the shared structures and cross-task patterns that are essential for robust performance on unseen or less frequent tasks (Lin et al., 2024b; Chen et al., 2024).

Inspired by the principles of Meta-Learning, which emphasizes enabling models to "learn how to learn" (Finn et al., 2017), we propose a novel framework called META-LORA, specifically designed

to address the data efficiency challenge in the fine-tuning process of LLMs within multi-task learning scenarios. To achieve this, we frame the fine-tuning process as an iterative optimization procedure consisting of two key stages: task-specific adaptation and meta-knowledge update. The task-specific adaptation stage allows for rapid per-task learning using only a small amount of data from the support set, enabling efficient adaptation to each task without requiring large datasets. Meanwhile, the meta-knowledge update stage aggregates insights from multiple tasks through a shared LoRA adapter, promoting the transfer of knowledge across tasks while minimizing data usage. Together, these two stages optimize the learning process, significantly enhancing data efficiency while preserving model performance. We demonstrate that META-LORA achieves competitive performance in both multi-task learning and multilingual learning scenarios while requiring significantly less task-specific data, showcasing its adaptability across a variety of tasks. Extensive ablation studies and analyses further demonstrate the necessity of the two-stage optimization framework.

In summary, the major contributions of this paper are outlined below.

- We propose META-LORA, a novel framework designed to enhance data efficiency for multi-task LoRA adaptation while preserving model performance.
- Comprehensive experimental evaluations validate the effectiveness of META-LORA in both multi-task learning and multilingual learning scenarios.
- Extensive ablation experiments and analyses demonstrate the necessity of the two-stage optimization framework.

## 2 RELATED WORK

**Parameter-Efficient Fine-tuning** As LLMs grow more powerful, fine-tuning them remains computationally intensive. This challenge has motivated the development of parameter-efficient fine-tuning (PEFT) methods, which aim to lower memory and storage demands during model adaptation. One representative approach is adapter tuning (Rebuffi et al., 2017b; Houlsby et al., 2019; Sung et al., 2022; Stickland & Murray, 2019), which introduces trainable layers into the existing model while keeping the original parameters frozen. Another line of PEFT research focuses on directly manipulating model activations through learnable vectors, with methods such as concatenation (Liu et al., 2024b; Li & Liang, 2021; Lester et al., 2021), multiplication, and addition. Additionally, prompt-based tuning methods like prefix tuning (Lester et al., 2021) and continuous prompt tuning (Li & Liang, 2021; Liu et al., 2021) replace discrete prompt engineering with trainable embeddings. Beyond injecting new parameters, researchers have also explored sparse updates (Sung et al., 2021; Dey et al., 2024) and low-rank adaptation (LoRA) (Hu et al., 2022) as alternatives that modify only a small subset of the model's parameters or its computational graph.

**Lora Architecture on Multi-Task Learning** Multi-LoRA Architectures have emerged as a promising solution for adapting LLMs, such as LLaMA, in resource-constrained settings (Touvron et al., 2023). To further leverage the potential of LoRA, researchers have proposed multi-LoRA approaches that employ multiple low-rank adapters simultaneously. For instance, MultiLoRA (Wang et al., 2023b) focuses on scaling LoRA modules horizontally and modifying parameter initialization to reduce parameter dependency. A particularly prominent direction involves integrating the Mixture-of-Expert (MoE) framework and PEFT, where multiple adapters are dynamically combined. For example, LoRAHub (Huang et al., 2023) trains multiple adapters and dynamically selects suitable combinations based on the domain at inference time. LoRAMoE (Dou et al., 2023) also incorporated an MoE framework to protect pre-trained knowledge during instruction tuning. More advanced MoE methods include MoLoRA (Zadouri et al., 2023), which trains multiple LoRA experts concurrently under a learned gating mechanism, and C-Poly (Wang et al., 2023a), which jointly learns a skill assignment matrix to combine task-common and task-specific skills. Furthermore, MoDULA (Ma et al., 2024) introduces a novel PEFT MoE paradigm that separates universal and domain-specific experts, training them in a three-stage process to achieve flexible pluggability.

**Data-efficient strategies on LLM** Fine-tuning LLMs is computationally demanding, which has motivated the development of data-efficient strategies to reduce resource consumption without sacrificing performance (Ding et al., 2023; Xu et al., 2024). Among these, two prominent approaches are coreset selection (Lin et al., 2024a; Xia et al., 2022; 2024) and data pruning (Marion et al., 2023;

Sorscher et al., 2022). These techniques highlight the promise of intelligent data selection for alleviating the computational burden of LLM fine-tuning, thereby improving accessibility in resource-constrained settings. Specifically, coreset selection focuses on identifying a representative subset of training data that can approximate the performance achieved with the full dataset (Lin et al., 2024a; Xia et al., 2022; 2024), while data pruning aims to eliminate less informative or redundant samples to streamline training (Marion et al., 2023; Sorscher et al., 2022).

# 3 META-LORA

In this section, we introduce META-LORA, a data-efficient LoRA-based architecture for fine-tuning, as illustrated in Figure 1. The detailed fine-tuning procedure is provided in Algorithm 1.

## 3.1 PROBLEM FORMULATION

Given a model $f$ with pre-trained weights $W_0$ and a set of tasks $\mathcal{T} = \{\mathcal{T}_1, \mathcal{T}_2, \ldots, \mathcal{T}_N\}$, current LoRA-based methods for multi-task adaptation optimize:

$$\min_{\theta} \sum_{i=1}^{N} \mathbb{E}_{(x,y) \sim \mathcal{T}_i} [\mathcal{L}(f_{W_0 + \Delta\theta}(x), y)] \quad (1)$$

where $\theta = \{A, B\}$ denotes the shared LoRA adapter parameters across all $N$ tasks, and $\Delta\theta = BA$ approximates the accumulated gradient updates $\Delta W_0$. $\mathcal{T}_i$ denotes the data distribution of the $i$-th task. Different from the above paradigm, we propose a two-stage optimization framework that maintains multiple local LoRA adapters during the task-specific adaptation stage, alongside a global LoRA adapter updated in the meta-knowledge update stage, enabling the model to efficiently adapt to multiple tasks while minimizing the amount of data required for each task.

## 3.2 FINE-TUNING

### 3.2.1 PHASE I: TASK-SPECIFIC ADAPTATION

In each iteration, we randomly sample $n$ tasks $\mathcal{B} = \{\mathcal{T}_1, \mathcal{T}_2, \ldots, \mathcal{T}_n\}$ from $\mathcal{T}$ with each task following the episodic formulation:

$$\mathcal{T}_i := (\mathcal{S}_i, \mathcal{Q}_i) \quad (2)$$

where $\mathcal{S}_i$ and $\mathcal{Q}_i$ are disjoint subsets randomly selected from task $\mathcal{T}_i$, traditionally referred to as the support set and the query set (Finn et al., 2017) with sizes $n_i^s$ and $n_i^q$, respectively.

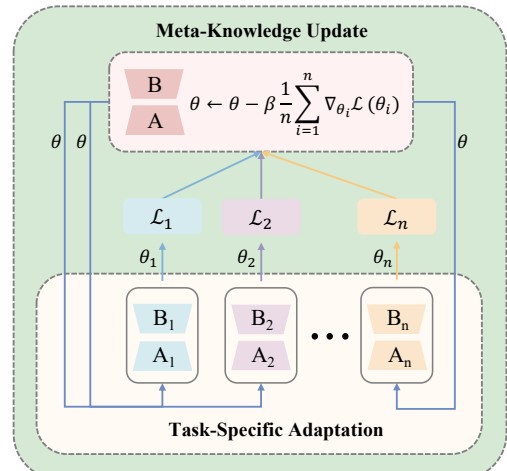

Figure 1: **Architecture of META-LORA.** During fine-tuning, META-LORA adopts a two-stage optimization framework comprising a task-specific adaptation stage and a meta-knowledge update stage. Starting from the initial shared LoRA parameters $\theta$, task-specific adapters are rapidly adapted using support set of each task, enabling efficient task-level specialization. In the second stage, the updated task-specific parameters $\theta_1, \theta_2, \ldots, \theta_n$ are used to compute gradients from the corresponding query sets, which are then aggregated to update the shared LoRA adapter, enabling effective cross-task knowledge transfer. By iteratively alternating between these two stages, META-LORA enhances data efficiency in multi-task adaptation of LLMs.

For each task $\mathcal{T}_i \in \mathcal{B}$, the task-specific adapter parameters $\theta_i$ are first initialized with the shared adapter parameters $\theta$ (i.e., $\theta_i \leftarrow \theta$). Subsequently, the model performs the gradient descent on $\theta_i$ for $k$ steps based on its corresponding support set, simulating the model's rapid adaptation to a new task. One gradient update of task-specific (local) LoRA adapter parameters $\theta_i$ can be formulated as:

$$\theta_i \leftarrow \theta_i - \alpha \nabla_{\theta_i} \mathcal{L}_{\mathcal{S}_i}(\theta_i) \quad (3)$$

where $\alpha$ is the learning rate of the adaptation stage, and the task-specific adaptation loss $\mathcal{L}_{\mathcal{S}_i}(\theta_i)$ is:

$$\mathcal{L}_{\mathcal{S}_i}(\theta_i) = \frac{1}{n_i^s} \sum_{(x,y) \sim \mathcal{S}_i} \mathcal{L}(f_{W_0 + \Delta\theta_i}(x), y) \quad (4)$$

---

**Algorithm 1** META-LORA using the First-Order Approximation

---

1: **Require:** Task set $\mathcal{T} = \{\mathcal{T}_1, \mathcal{T}_2, \ldots, \mathcal{T}_N\}$, model $f$ with frozen pre-trained parameters $W_0$, learning rate of the adaptation stage $\alpha$, learning rate of the meta-update stage $\beta$, number of adaptation steps $k$, number of selected tasks in each iteration $n$

2: Randomly initialize the shared LoRA adapter parameters $\theta$

3: **for** each iteration **do**

4:     **Stage I: Task-Specific Adaptation**

5:     Randomly select $n$ tasks $\mathcal{B} \subseteq \mathcal{T}$

6:     **for** each task $\mathcal{T}_i \in \mathcal{B}$ **do**

7:         Sample support set $\mathcal{S}_i$ and query set $\mathcal{Q}_i$

8:         Create a temporary copy of parameters for adaptation: $\theta_i \leftarrow \theta$

9:         Obtain task-specific adapted parameters $\theta_i$ by performing Eq. 3 for $k$ steps

10:     **end for**

11:     **Stage II: Meta-Knowledge Update**

12:     Compute the generalization loss for each selected task $\mathcal{T}_i$ using Eq. 6

13:     Average the generalization gradients to update the shared LoRA parameters $\theta$ using Eq. 5

14: **end for**

15: **Output:** Fine-tuned LoRA parameters $\theta$

---

### 3.2.2 PHASE II: META-KNOWLEDGE UPDATE

After obtaining the adapted parameters $\theta_i$ for each task $\mathcal{T}_i$, the meta-update is performed on the shared (global) LoRA parameters $\theta$. To maintain computationally tractable with large models, we employ the first-order approximation of MAML, a common simplification that retains meta-learning signals while avoiding the prohibitive computational overhead of Hessian-vector products. In practice, this is implemented by detaching the computational graph of the task-specific adaptation, which prevents backpropagation through the task-specific update steps. The approximate meta-update rule, which averages the gradients from a batch of $n$ sampled tasks, is therefore formulated as:

$$\theta \leftarrow \theta - \beta \frac{1}{n} \sum_{i=1}^{n} \nabla_{\theta_i} \mathcal{L}_{\mathcal{Q}_i}(\theta_i) \tag{5}$$

where $\beta$ is the learning rate in the meta-stage, and the generalization loss $\mathcal{L}_{\mathcal{Q}_i}(\theta_i)$ with respect to task $\mathcal{T}_i$ is computed on the corresponding query set using the adapted LoRA parameters $\theta_i$:

$$\mathcal{L}_{\mathcal{Q}_i}(\theta_i) = \frac{1}{n_i^q} \sum_{(x,y) \sim \mathcal{Q}_i} \mathcal{L}(f_{W_0 + \Delta\theta_i}(x), y) \tag{6}$$

### 3.3 INFERENCE

After fine-tuning, we obtain a single shared adapter that captures multi-task knowledge. It is worth emphasizing that the task-specific adapters are merely intermediate artifacts generated during the fine-tuning process. In the inference stage, the parameters of the shared adapter $\theta$ are seamlessly merged into the frozen pre-trained weights $W_0$, eliminating the need to activate or switch among task-specific adapters.

## 4 EXPERIMENTS

In this section, we present a series of experiments to evaluate the effectiveness of META-LORA in both multi-task and multilingual learning scenarios. Additionally, we conduct ablation studies to examine the necessity of the proposed two-stage optimization framework, as well as the adaptability of META-LORA under more challenging multi-task configuration. Finally, we perform a sensitivity analysis to investigate the impact of the amount of fine-tuning data on overall performance.

### 4.1 EXPERIMENT SETTING

#### 4.1.1 MULTI-TASK LEARNING

In multi-task learning scenario, we consider two task configurations. First, we adopt the same experimental setting described in HydraLoRA (Tian et al., 2024). Additionally, we develop a five-task dataset comprising semantically and structurally diverse samples, to further evaluate the model's capacity to handle varied tasks.

*Flanv2 Setting*

**Dataset.** We fine-tune the model using a subset of the Flanv2 dataset (Wei et al., 2021) that includes 43 tasks from both Natural Language Understanding (NLU) and Natural Language Generation (NLG), grouped into 10 distinct task clusters. The model's performance is evaluated using the Big-Bench Hard (BBH) (Suzgun et al., 2022) benchmark. More details of the dataset can be found in Appendix A.1.1.

**Baselines.** We evaluate MeTA-LoRA against other LoRA-based methods designed for multi-task learning across multiple datasets: (1) Lorahub (Huang et al., 2023), which utilizes black-box optimization to learn weights for 20 randomly selected LoRAs for new tasks, applying weighted averaging without the need for gradient calculations; (2) LoRA MoE (Liu et al., 2024a), which combines lightweight experts (LoRA) with a Mixture of Experts (MoE) architecture for high efficiency, enabling generalization to new tasks without prior knowledge; (3) HydraLoRA (Tian et al., 2024), which employs Multi-Head structure in conjunction with MoE to achieve a balance between parameter efficiency and training effectiveness; (4) R-LoRA (Liu et al., 2025a), which utilizes random initialization of multi-head LoRA for multi-task learning.

*Five-task Setting*

**Dataset.** We build a five-task dataset encompassing GSM8K (Cobbe et al., 2021) for arithmetic reasoning, QQP (Wang et al., 2017) and CosmosQA (Huang et al., 2019) for NLU, SiQA (Sap et al., 2019) and PiQA (Bisk et al., 2020) for commonsense reasoning. Then, we use the MMLU (Massive Multitask Language Understanding) (Hendrycks et al., 2020) benchmark to measure the world knowledge acquired during fine-tuning and problem solving ability of models. In addition, we include the BBH benchmark. Following common practice, we conduct 5-shot evaluation on MMLU and 3-shot evaluation on BBH.

**Baselines.** Following prior studies (Tian et al., 2024; Wang et al., 2025), we adopt LLaMA2-7B and LLaMA2-13B as the backbone models. For comparison, we evaluate META-LORA against two representative baselines: (1) LoRA, a widely used approach for parameter-efficient fine-tuning; (2) HydraLoRA, a representative method tailored for multi-task learning.

#### 4.1.2 MULTILINGUAL LEARNING

In multilingual learning scenario, we fine-tune the LLaMA2-7B and LLaMA2-13B using Bactrian-X (Li et al., 2023), a comprehensive multilingual parallel dataset comprising 3.4 million instruction–response pairs across 52 languages. Bactrian-X is automatically constructed by translating instructions from Alpaca (Taori et al., 2023) and Dolly (Conover et al., 2023) via the Google Translate API.2. In our evaluation, we compare META-LORA against two baselines: (1) the corresponding vanilla models; (2) the multilingual Bactrian models (BX), which are fine-tuned on the full Bactrian-X dataset. To probe the zero-shot language understanding capability of the different models and how much knowledge of they encode, we evaluate on the following four benchmarks:

- XCOPA (Ponti et al., 2020): a multilingual resource designed for causal commonsense reasoning, encompassing 11 languages from 11 families and several areas around the globe. The task requires selecting the correct subsequent sentence from two given options, based on cause and effect question types.

- XStoryCloze (Lin et al., 2022): the professionally translated version of the English StoryCloze dataset (Mostafazadeh et al., 2016) into 10 non-English languages. The task involves selecting one sentence as a plausible ending (closure) from two options, given a four-sentence story as the premise.

Table 1: Performance of different methods on BBH (3-shot) with LLaMA2-7B and LLaMA2-13B fine-tuned on the subset of the Flanv2 dataset. META-LORA uses only 100 examples sampled from each task for fine-tuning, whereas the other methods except Base utilize the full dataset. * indicates results from Liu et al. (2025a).

| Model Size | Base | LoRA | LoRAHub* | LoRA MoE* | HydraLoRA* | R-LoRA* | META-LORA |
|---|---|---|---|---|---|---|---|
| 7B | 31.6 | 37.2 | 39.7 | 40.3 | 41.8 | **42.2** | 38.52 |
| 13B | 38.4 | 40.9 | 41.9 | 43.7 | 44.7 | 45.1 | **46.32** |

Table 2: Performance comparison on the proposed five-task dataset. The base models are fine-tuned by META-LORA with only 50 examples randomly sampled from each task, and LoRA and HydraLoRA represent the full-data tuning schemes.

| Method | MMLU | MMLU-math | BBH | AVG |
|---|---|---|---|---|
| | | LLaMA2-7B | | |
| LoRA | 29.43 | 26.42 | 30.24 | 28.70 |
| HydraLoRA | 46.61 | 30.00 | 37.49 | 38.03 |
| META-LORA | 46.34 | 31.12 | 39.53 | **39.00** |
| | | LLaMA2-13B | | |
| LoRA | 39.99 | 26.08 | 44.84 | 36.97 |
| HydraLoRA | 54.24 | 30.88 | 44.67 | 43.26 |
| META-LORA | 54.18 | 30.97 | 46.12 | **43.76** |

- XWinoGrad (Tikhonov & Ryabinin, 2021; Muennighoff et al., 2022): a multilingual assessment dataset for commonsense reasoning, made up of Winograd Schema Challenge problems in six languages. The objective is to select the most plausible sentence from options that differ slightly.

- EXAMS (Hardalov et al., 2020): A multilingual multiple-choice question-answering dataset constructed from high school exam questions in 16 languages, covering a wide range of subjects including natural sciences (e.g., physics), social sciences (e.g., history), and humanities (e.g., philosophy).

### 4.1.3 HYPER-PARAMETER SETTINGS

For both multi-task learning and multilingual learning scenarios, we run META-LORA for 1,000 iterations. In each iteration, we randomly select 2 tasks (i.e. $n = 2$) for adaptation. For each task $i$, the size of the support set $n_i^s$ and the query set $n_i^q$ are both set to 8. We perform 3 steps of gradient descent on the support set to obtain task-specific LoRA adapters, using a learning rate of $5 \times 10^{-6}$.

Table 3: Statistics of different settings with respect to the number of tasks and the amount of samples used by the baselines and the proposed META-LORA.

| Settings | Tasks | Baselines | Ours |
|---|---|---|---|
| Flanv2 | 43 | 325,783 | 4,300 |
| Five-task | 5 | 439,950 | 250 |
| Multilingual | 52 | 3,400,000 | 2,600 |
| Five-task variant | 5 | 487,075 | 250 |

During the meta-knowledge update phase, we employ AdamW optimizer with a learning rate of $2 \times 10^{-6}$. A comprehensive comparison of the number of fine-tuning samples used by the baselines and our method under these settings is presented in Table 3, and additional details regarding the hyper-parameter settings can be found in Appendix A.1.2.

Table 4: Averaged zero-shot accuracy for XCOPA, XStoryCloze, XWinograd, and EXAMS under different tuning schemes. META-LORA fine-tunes the base models with only 50 examples randomly sampled from each language, and $BX_{LLaMA}$ represents LoRA tuning on the full Bactrian-X dataset. * indicates results from Li et al. (2023).

| Model | XCOPA | XStoryCloze | XWinograd | EXAMS | AVG |
|-------|-------|-------------|-----------|-------|-----|
| LLaMA* (7B) | 50.22 | 57.03 | 57.96 | 28.20 | 48.35 |
| $BX_{LLaMA}$* (7B) | 51.76 | **58.91** | 60.16 | **29.14** | 49.99 |
| **META-LORA (7B)** | **57.02** | 58.46 | **79.12** | 23.84 | **54.61** |
| LLaMA* (13B) | 51.04 | 57.88 | 52.97 | 30.41 | 48.08 |
| $BX_{LLaMA}$* (13B) | 53.27 | **62.12** | 63.65 | **35.71** | 53.69 |
| **META-LORA (13B)** | **57.87** | 59.58 | **82.18** | 23.84 | **55.87** |

Table 5: Performance comparison on the more challenging five-task dataset, where PiQA is replaced by WinoGrande. META-LORA fine-tunes LLaMA2-7B and LLaMA2-13B using 50 examples per task, while LoRA and HydraLoRA correspond to fine-tuning on the entire datatset.

| Method | MMLU | MMLU-math | BBH | AVG |
|--------|------|-----------|-----|-----|
| LLaMA2-7B | | | | |
| LoRA | 39.03 | 29.16 | 32.01 | 33.40 |
| HydraLoRA | 47.19 | 30.98 | 36.34 | 38.17 |
| META-LORA | 46.50 | 32.15 | 39.07 | **39.24** |
| LLaMA2-13B | | | | |
| LoRA | 39.17 | 28.06 | 43.44 | 36.89 |
| HydraLoRA | 54.12 | 30.41 | 44.28 | 42.94 |
| META-LORA | 54.18 | 30.51 | 46.02 | **43.57** |

## 4.2 PERFORMANCE

### 4.2.1 PERFORMANCE ON MULTI-TASK LEARNING

Table 1 and Table 2 present the performance of various fine-tuned models evaluated on the different benchmarks, using a subset of the Flanv2 dataset and the proposed five-task dataset, respectively. We make several observations in more detail, and discuss them below.

- As shown in Table 1, META-LORA consistently improves performance over both the base models and the models fine-tuned with standard LoRA. Notably, based on the LLaMA2-13B model, META-LORA achieves the highest BBH score of 46.32 among all LoRA-based baselines, despite involving only 100 examples per task into the fine-tuning process. The improvements highlight the effectiveness and data efficiency of the META-LORA framework.

- When a more diverse set of tasks is used for fine-tuning, as shown in Table 2, META-LORA consistently outperforms both standard LoRA and HydraLoRA on LLaMA2-7B and LLaMA2-13B, even with only 50 examples per task. Overall, META-LORA surpasses HydraLoRA and LoRA by 0.97% and 10.3% on LLaMA2-7B, and by 0.50% and 6.79% on LLaMA2-13B, respectively, indicating that it can effectively capture task-specific knowledge and thereby enhances generalization across a broad range of tasks.

### 4.2.2 PERFORMANCE ON MULTILINGUAL LEARNING

The average performance across all languages for XCOPA, XStoryCloze, XWinograd, and EX-AMS is reported in Table 4. Notably, even with only 50 randomly sampled examples per language, META-LORA achieves competitive or superior performance compared to the BX models, particularly on XCOPA and XWinograd. Moreover, it significantly outperforms the vanilla LLaMA2 models without multilingual adaptation on three out of four benchmarks. This substantial improvement

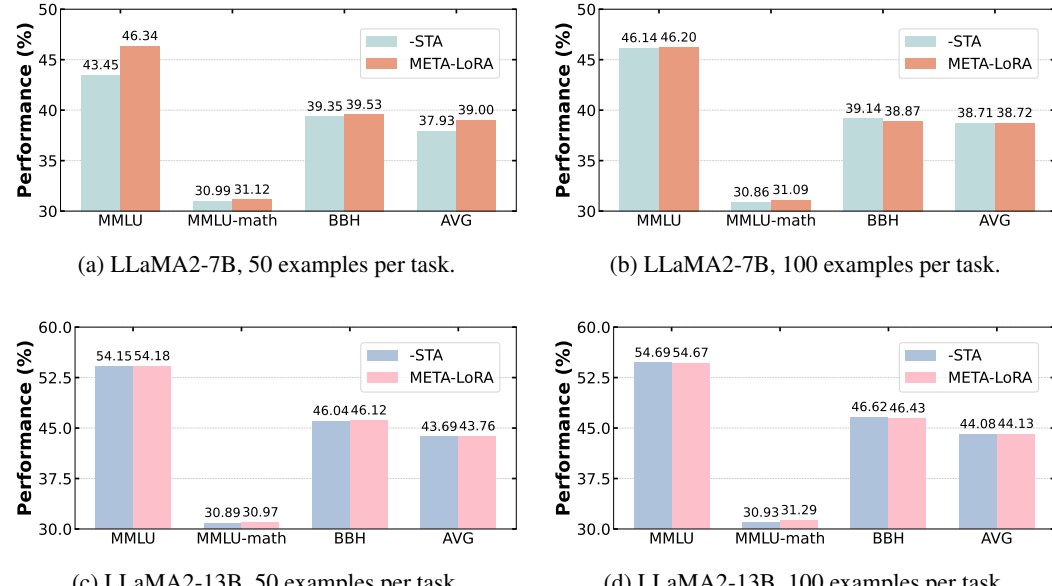

Figure 2: Ablation results on LLaMA2-7B and LLaMA2-13B under the five-task learning setting. **-STA** refers to the variant without the task-specific adaptation stage, highlighting its contribution to overall performance.

in language understanding can be attributed to the META-LORA mechanism, which enables rapid adaptation across diverse languages using only a limited amount of data. By leveraging its two-stage optimization framework, META-LORA effectively captures both shared multilingual knowledge and language-specific patterns, thereby achieving strong data efficiency in multilingual adaptation.

## 4.3 ABLATION STUDIES

### RQ1: Do more challenging datasets better reveal the generation ability of models?

To further assess the effectiveness of META-LORA in more challenging multi-task learning scenarios, we construct a new five-task configuration by replacing PiQA with WinoGrande (Sakaguchi et al., 2021). Compared to PiQA, WinoGrande is a larger and more difficult commonsense reasoning dataset. It features cloze-style questions in which the model is required to select the correct option from two candidates to complete a given sentence. We adopt LLaMA2-7B and LLaMA2-13B as base models. LoRA and HydraLoRA serve as the baselines by fine-tuning the base models on the full dataset, while META-LORA uses only 50 examples per task. The evaluation results on MMLU, MMLU-math, and BBH are summarized in Table 5.

It can be observed that, compared to the standard five-task configuration (as shown in Table 2), fine-tuning LLaMA2-7B on the more challenging task sets using LoRA, HydraLoRA or META-LORA leads to a moderate improvement in the reasoning ability of the model. Specifically, LoRA, HydraLoRA and META-LORA achieve average performance gains of 4.70%, 0.14% and 0.24%, respectively. For LLaMA2-13B, although overall performance tends to saturate due to the strong inherent generalization of the model, META-LORA continues to substantially outperform LoRA and HydraLoRA across configurations, highlighting the robustness of META-LORA under more challenging task distributions and its scalability to larger models.

### RQ2: Is the two-stage optimization framework necessary for the fine-tuning process?

To further understand the contribution of the task-specific adaptation stage in META-LORA, we conduct an ablation study, with the results presented in Figure 2. Specifically, we investigate a variant, denoted as -STA, in which the task-specific adaptation stage is removed. In this variant, only the gradients computed from the query sets of multiple selected tasks are aggregated to update the shared LoRA adapters in each iteration. Experiments are performed using two backbone models,

Table 6: Comparison of LoRA and META-LORA on the standard five-task learning scenario with varying amounts of fine-tuning data per task.

| # Data / task | Method | MMLU | MMLU-math | BBH | AVG |
|---|---|---|---|---|---|
| | | LLaMA2-7B | | | |
| 50 | LoRA | 46.60 | 32.60 | 31.06 | 36.75 |
| | HydraLoRA | 46.21 | 29.42 | 37.30 | 37.64 |
| | META-LORA | 46.34 | 31.12 | 39.53 | **39.00** |
| 100 | LoRA | 46.49 | 27.32 | 37.17 | 36.99 |
| | HydraLoRA | 46.72 | 29.78 | 37.44 | 37.98 |
| | META-LORA | 46.20 | 31.09 | 38.87 | **38.72** |
| | | LLaMA2-13B | | | |
| 50 | LoRA | 54.17 | 30.47 | 45.89 | 43.51 |
| | HydraLoRA | 52.82 | 31.22 | 34.80 | 39.61 |
| | META-LORA | 54.18 | 30.97 | 46.12 | **43.76** |
| 100 | LoRA | 55.12 | 31.11 | 45.96 | 44.06 |
| | HydraLoRA | 54.11 | 31.30 | 40.50 | 41.97 |
| | META-LORA | 54.67 | 31.29 | 46.43 | **44.13** |

LLaMA2-7B and LLaMA2-13B, under the proposed five-task learning configuration. For each task, either 50 or 100 examples are randomly selected for fine-tuning. In addition, both -STA and META-LORA are evaluated on three benchmarks: MMLU, the mathematics tasks of MMLU and BBH.

As shown in Figure 2, the task-specific adaptation stage contributes positively to model performance across both model scales and data regimes, illustrating the necessity and effectiveness of the two-stage design. Notably, the performance gains introduced by the task-specific adaptation stage are more pronounced when using the smaller model (LLaMA2-7B) and the more limited amount of fine-tuning data (50 examples per task). In this case, META-LORA improves the average score by 1.07%, demonstrating its ability to capture more task-specific knowledge prior to meta-aggregation. Moreover, while the larger LLaMA2-13B model already exhibits strong generalization capabilities even with a simplified optimization process, the two-stage structure continues to deliver consistent performance benefits, confirming its robustness and scalability across model sizes.

## 4.4 PARAMETER ANALYSIS

**RQ3: Does scaling the fine-tuning data always improve performance in multi-task learning?**

As shown in Table 6, we analyze how the scale of fine-tuning data impacts model performance under the standard five-task learning setting. Specifically, we fine-tune LLaMA2-7B and LLaMA2-13B using LoRA, HydraLoRA and META-LORA with either 50 or 100 examples per task. Subsequently, we evaluate the overall performance on MMLU, MMLU-math and BBH. Additional results on the more challenging variant of this setting are reported in Table 12 of Appendix A.3.5.

Several insights can be drawn from the results. Overall, increasing the number of fine-tuning examples per task generally improves performance. Moreover, META-LORA consistently achieves higher average scores across model sizes and data regimes compared to both LoRA and HydraLoRA. For example, under the standard five-task setting with only 50 examples per task, META-LORA surpasses LoRA and HydraLoRA on LLaMA2-7B by 2.25% and 1.36% in average score, respectively. This advantage stems from its novel two-stage optimization framework, which enables strong gains in low-resource settings by extracting task-specific knowledge from only a few examples, while also maintaining stable improvements at larger data scales by mitigating task conflicts.

Interestingly, LoRA sometimes performs better with only 50 or 100 examples per task than with the full dataset. This counter-intuitive result arises from architectural limitations in multi-task learning. LoRA uses a single shared low-rank adapter that struggles to capture the heterogeneous requirements of diverse tasks. As the data scale grows, task-specific signals interfere with one another, leading to degraded performance. HydraLoRA alleviates this issue to some extent with its asymmetric design.

## 5 CONCLUSION

In this work, we introduce META-LORA, a simple yet effective two-stage optimization framework designed to enhance data efficiency in multi-task adaptation of LLMs. By explicitly decoupling task-specific adaptation and meta-knowledge aggregation, META-LORA is able to quickly adapt to individual tasks using only a few examples, while simultaneously promoting cross-task generalization through shared parameter updates. Comprehensive experiments across both multi-task and multilingual learning settings demonstrate META-LORA consistently matches or outperforms standard full-data LoRA fine-tuning, despite using significantly less task-specific supervision. These results highlight the potential of META-LORA as a practical and scalable solution for efficient fine-tuning in real-world low-resource and multi-task scenarios.

## 6 REPRODUCIBILITY STATEMENT

We place a strong emphasis on the transparency and reproducibility of our work. The main text provides comprehensive descriptions of the proposed method and the model architecture. Additional details regarding datasets, hyperparameter configurations, and extended experimental results are included in the appendix. To further facilitate verification and extension by the research community, we will release the complete source code for fine-tuning and inference along with the running scripts used to produce the experimental results reported in this paper upon publication.

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

# A APPENDIX

## A.1 DATASETS AND HYPER-PARAMETERS

### A.1.1 DETAILS FOR FLANV2 SETTING

Following (Tian et al., 2024), we select a portion of the Flanv2 datasets covering Natural Language Understanding (NLU) and Natural Language Generation (NLG), which can be grouped into 10 distinct task clusters. Then we evaluate it with the Big-Bench Hard (BBH) benchmark.

We summarize the details of the used datasets as follows:

1. **Struct-to-Text Conversion**: This task evaluates the capability to generate natural language descriptions from structured data inputs. We use the following datasets: (1) CommonGen; (2) DART; (3) E2ENLG; (4) WebNLG

2. **Translation**: Translation involves converting text from one language to another, maintaining the original meaning and nuances. We use the following datasets: (1) En-Fr from WMT'14; (2) En-De, En-Tr, En-Ru, En-Fi, En-Ro from WMT'16; (3) En-Es from Paracrawl.

3. **Commonsense Reasoning**: This involves assessing the ability to apply physical or scientific principles alongside common sense in reasoning tasks. We use the following datasets: (1) COPA; (2) HellaSwag; (3) PiQA; (4) StoryCloze.

4. **Sentiment Analysis**: A fundamental task in natural language processing (NLP) that determines the sentiment polarity (positive or negative) of a given text. We use the following datasets: (1) IMDB; (2) Sentiment140; (3) SST-2; (4) Yelp.

5. **Paraphrase Detection**: This task requires models to ascertain whether two sentences convey the same meaning, indicating semantic equivalence. We use the following datasets: (1) MRPC; (2) QQP; (3) Paws Wiki.

6. **Coreference Resolution**: Involves identifying instances within a text that refer to the same entity, demonstrating an understanding of textual context. We use the following datasets: (1) DPR; (2) WSC273.

7. **Reading Comprehension**: Assesses the capability to derive answers to questions from a provided text containing relevant information. We use the following datasets: (1) BoolQ; (2) DROP; (3) MultiRC; (4) OBQA; (5) SQuADv1; (6) SQuADv2.

8. **Reading Comprehension with Commonsense**: Merges traditional reading comprehension skills with commonsense reasoning, requiring understanding beyond the explicit text. We use the following datasets: (1) CosmosQA; (2) ReCoRD.

9. **Natural Language Inference**: Focuses on deducing the relationship between two sentences, determining if the second sentence logically follows from, contradicts, or is unrelated to the first sentence. We use the following datasets: (1) ANLI; (2) CB; (3) MNLI; (4) QNLI; (5) SNLI; (6) WNLI; (7) RTE.

10. **Closed-Book Question Answering**: This task challenges models to answer questions about general knowledge without direct access to external information sources. We use the following datasets: (1) ARC; (2) NQ; (3) TriviaQA.

### A.1.2 HYPER-PARAMETER SETTINGS

Details on hyperparameters used for META-LORA, LoRA and HydraLoRA are provided below.

**META-LORA**: For all experiments, we integrate adapter modules into every dense layer of the multi-head attention (namely $Q$, $K$, $V$, and $O$) in the selected LLMs. Also, we set the low-rank parameter $r$ to 16.

**LoRA**: In all experiments, adapter modules are inserted into every dense layer of the multi-head attention components (namely $Q$, $K$, $V$, and $O$) in the selected LLMs, with the rank $r$ set to 16. In addition, the learning rate is set to $2 \times 10^{-4}$ and the batch size is set to 8. For experiments on the full datasets, results are reported after a single epoch of fine-tuning. For parameter analysis with

Table 7: Comparison of efficiency across fine-tuning schemes. LoRA and HydraLoRA fine-tune the base models on the more challenging dataset for one epoch, while META-LORA performs 1,000 iterations using 100 samples from each task.

| Model Size | LoRA | HydraLoRA | META-LORA |
|---|---|---|---|
| Optimizer Steps | $\approx 60{,}885$ | $\approx 60{,}885$ | **1,000** |
| Total Samples Processed | 487,075 | 487,075 | **64,000** |
| Running Time (7B) | 17h | 37h | **3h** |
| Running Time (13B) | 29h | 57h | **5h** |

Table 8: Performance comparison on the standard five-task setting with respect to $\alpha$ and $\beta$.

| Combination | $\alpha$ | $\beta$ | BBH |
|---|---|---|---|
| $c_{base}$(ours) | $5 \times 10^{-6}$ | $2 \times 10^{-6}$ | **39.53** |
| $c_1$ | $2 \times 10^{-6}$ | $5 \times 10^{-6}$ | 39.39 |
| $c_2$ | $5 \times 10^{-5}$ | $2 \times 10^{-5}$ | 37.90 |
| $c_3$ | $5 \times 10^{-7}$ | $2 \times 10^{-7}$ | 38.96 |

respect to the amount of fine-tuning data, LoRA fine-tunes the base models for 5 epochs when using 50 examples per task, and for 10 epochs when using 100 examples per task. Finally, we report the best results obtained on the benchmarks.

**HydraLoRA**: For experiments not covered in the original paper (Tian et al., 2024), we adopt the default configurations suggested by HydraLoRA and adjust the number of $B$ matrices (`lora_nums`) to match the number of tasks. Consistent with LoRA, full-dataset fine-tuning is performed for a single epoch. Under limited-data settings, the base models are fine-tuned for 5 epochs with 50 examples per task and for 10 epochs with 100 examples per task. Finally, we report the best results obtained in each setting.

## A.2 TRAINING EFFICIENCY

To further assess the efficiency of the proposed method, we provide a detailed breakdown of the computational cost in Table 7. We report **Total Samples Processed** as a high-fidelity proxy for total FLOPs and token counts, given that the average sequence length is consistent across all methods. As shown in the Table 7, the baselines process the full dataset of 487,075 samples and require $\approx$60,885 optimizer steps. In contrast, META-LORA operates on a meta-learning objective over 1,000 iterations. Crucially, due to the sparse sampling mechanism in its inner and outer loops, META-LORA processes a total of only 64,000 samples. Consequently, our method achieves competitive performance while incurring only 13.1% of the total computational cost and 1.6% of the optimizer steps required by the baselines. In addition, META-LORA substantially reduces training time compared to standard LoRA and advanced HydraLoRA, while maintaining competitive performance.

## A.3 PARAMETER ANALYSIS

### A.3.1 IMPACT OF LEARNING RATES

To investigate the impact of learning rates, we fine-tuned LLaMA2-7B under the standard five-task learning setting using various $(\alpha, \beta)$ configurations. As presented in Table 8, $c_2$ with higher learning rates and $c_3$ with lower learning rates both degrade overall performance, suggesting optimization instability and insufficient adaptation.

Table 9: Performance comparison on the standard five-task setting with respect to the number of sampled tasks $n$.

| $n$ | 1 | 2 (ours) | 3 |
|---|---|---|---|
| BBH | 38.89 | **39.53** | 39.43 |

Table 10: Performance comparison on the standard five-task setting with respect to the number of adaptation steps $k$.

| $k$ | 1 | 2 | 3 (ours) | 4 |
|---|---|---|---|---|
| BBH | 39.44 | 39.25 | **39.53** | 39.17 |

Table 11: Performance comparison on the standard five-task setting with respect to the support and query set sizes $(n_s, n_q)$.

| Combination | $n_s$ | $n_q$ | BBH |
|---|---|---|---|
| $s_{base}$(ours) | 8 | 8 | **39.53** |
| $s_1$ | 2 | 2 | 39.18 |
| $s_2$ | 4 | 4 | 38.82 |

### A.3.2 IMPACT OF THE NUMBER OF SAMPLED TASKS

As shown in Table 9, varying $n$ from 1 to 3 reveals that $n = 2$ yields the optimal performance. With $n = 1$, the meta-update relies on the gradient of a single task, likel leading to high variance and unstable optimization. Increasing $n$ to 2 enhances gradient stability by averaging across tasks. However, further increasing $n$ to 3 results in a slight performance drop, potentially due to conflicting optimization directions arising from the aggregation of too many diverse tasks.

### A.3.3 IMPACT OF THE NUMBER OF ADAPTATION STEPS

Table 10 examines the impact of varying the inner-loop steps $k$, identifying $k = 3$ as the optimal configuration. Fewer steps ($k = 1, 2$) result in under-adaptation, failing to adequately capture task-specific distributions. Conversely, increasing $k$ to 4 induces performance degradation. This aligns with established meta-learning findings: excessive updates in few-shot regimes prone the model to overfitting the support set, thereby compromising generalization to the query set.

### A.3.4 IMPACT OF THE SIZES OF THE SUPPORT AND QUERY SETS

Finally, we examined the impact of varying the sizes of support and query sets, as detailed in Table 11. Our default configuration ($n_s = n_q = 8$) demonstrates superior performance compared to smaller set sizes. This suggests that extremely small support sets induce high variance in gradient estimation, failing to provide reliable directions for task adaptation. In contrast, a moderate size of 8 strikes an optimal balance between computational efficiency and the fidelity of gradient signals.

### A.3.5 IMPACT OF THE FINE-TUNING DATA

As shown in Table 12, we also analyze how the scale of fine-tuning data impacts model performance under the more challenging variant of the standard five-task setting. Specifically, we fine-tune LLaMA2-7B and LLaMA2-13B using LoRA, HydraLoRA and META-LORA with either 50 or 100

Table 12: Comparative analysis of LoRA, HydraLoRA and META-LORA on the more challenging five-task learning scenario with varying amounts of fine-tuning data per task.

| # Data / task | Method | MMLU | MMLU-math | BBH | AVG |
|---|---|---|---|---|---|
| | LLaMA2-7B | | | | |
| 50 | LoRA | 47.04 | 31.32 | 34.90 | 37.75 |
| | HydraLoRA | 46.30 | 30.17 | 36.64 | 37.70 |
| | META-LORA | 46.50 | 32.15 | 39.07 | **39.24** |
| 100 | LoRA | 45.68 | 25.94 | 33.75 | 35.12 |
| | HydraLoRA | 46.44 | 30.51 | 36.75 | 37.90 |
| | META-LORA | 46.55 | 31.59 | 39.45 | **39.20** |
| | LLaMA2-13B | | | | |
| 50 | LoRA | 54.14 | 31.69 | 46.18 | **44.00** |
| | HydraLoRA | 53.18 | 31.83 | 36.25 | 40.42 |
| | META-LORA | 54.18 | 30.51 | 46.02 | 43.57 |
| 100 | LoRA | 54.36 | 29.22 | 46.17 | 43.25 |
| | HydraLoRA | 53.43 | 30.90 | 38.35 | 40.89 |
| | META-LORA | 54.23 | 31.21 | 46.78 | **44.07** |

Table 13: Performance comparison of LoRA and META-LORA across three data sampling strategies: Random sampling, BM25, and DSIR.

| # Data / task | Method | MMLU | MMLU-math | BBH | AVG |
|---|---|---|---|---|---|
| | LLaMA2-7B | | | | |
| 50 | LoRA + Rand | 46.60 | 32.60 | 31.06 | 36.75 |
| | LoRA + BM25 | 46.74 | 31.72 | 30.70 | 36.39 |
| | LoRA + DSIR | 46.83 | 31.63 | 30.22 | 36.23 |
| | META-LORA + Rand | 46.34 | 31.12 | 39.53 | **39.00** |

examples per task. Subsequently, we evaluate the overall performance on MMLU, MMLU-math, and BBH.

Consistent with the findings under the standard five-task setting, LoRA achieves better performance with limited data than with full-data, reflecting its inherent architectural limitations. When the difficulty of tasks increases, HydraLoRA struggles to capture task-specific knowledge in such low-resource regimes. In contrast, our method built upon a two-stage optimization framework can rapidly adapt in diverse multi-task scenarios, which further demonstrates its robustness and establishes its superiority over existing approaches.

## A.4 COMPARISON WITH DATA-EFFICIENT STRATEGIES

To more directly demonstrate the advantages of the proposed approach and resolve the conceptual gap between optimization and data selection, we conducted new comparative experiments against two established data-selection heuristics: (1) BM25 (Robertson et al., 2009), a statistical retrieval approach utilizing TF-IDF ranking; (2) DSIR (Xie et al., 2023), a feature-based weighting approach leveraging n-gram features. Specially, under the standard five-task setting, we first curated a subset of top-50 examples from each task based on their calculated influence scores for the BBH benchmark, which aligns with the stringent low-resource constraint of our method. Subsequently, we fine-tuned the LLaMA2-7B backbone using LoRA on the curated data, and then evaluated the performance across the BBH, MMLU, and MMLU-math benchmarks. As shown in Table 13, META-LORA achieves superior performance compared to LoRA using these highly curated data subsets. This confirms that the advantage of our framework is optimization-centric, specifically its algorithmic ability to leverage scarce data, rather than reliance on data selection heuristics.

### A.5 THE USAGE OF LARGE LANGUAGE MODELS

Large Language Models (LLMs) were employed exclusively as writing assistants to assist in the phrasing, clarity, and stylistic polishing of the manuscript. Their role was limited to improving the readability and presentation, without any involvement in research ideation, experimental design, analysis of results, or method development. All scientific contributions were conceived and carried out entirely by the authors, including mathematical formulations, algorithmic design, and empirical validation.

