# OpenReview forum: "MeTA-LoRA: Data-Efficient Multi-Task Fine-Tuning for Large Language Models"
_ICLR.cc/2026/Conference — Submitted to ICLR 2026_

### Official Review · Reviewer_r5pv · 2025-10-30

**Soundness:** 2
**Presentation:** 2
**Contribution:** 3
**Rating:** 4
**Confidence:** 4

**Summary:**

This paper proposes a novel framework named META-LORA, designed to enhance the data efficiency of large language models during LoRA fine-tuning in multi-task scenarios through a two-stage meta-learning approach. The method first rapidly acquires task knowledge on a small number of samples through a Task-Specific Adaptation phase, followed by a Meta-Knowledge Update phase that aggregates gradients across multiple tasks to promote knowledge generalization. Experimental results demonstrate that META-LORA achieves performance comparable to or surpassing baseline methods fine-tuned on full datasets across multi-task and multilingual benchmarks, using only a minimal number of unique samples.

While the problem addressed is highly significant and the proposed approach innovative, its core conclusions rest on experimental comparisons that warrant further scrutiny. Furthermore, the method's complexity and robustness are not sufficiently validated through ablation experiments, leaving its advantages and applicability unclear.

**Strengths:**

1. This work addresses the “data efficiency” challenge in fine-tuning large models for multi-task scenarios. This represents a widespread and critical challenge in real-world applications, where acquiring large volumes of high-quality labeled data is often prohibitively expensive.


2. The integration of meta-learning's “learning how to learn” philosophy with the LoRA framework constitutes a novel and intelligent approach. Decoupling the fine-tuning process into two distinct phases: task adaptation and meta-knowledge updating, is theoretically coherent and logically sound, offering fresh perspectives for addressing knowledge transfer challenges in multi-task learning.

3. META-LORA ultimately produces a single, shared LoRA adapter. This enables highly efficient model deployment and inference without requiring switching or combining multiple adapters for different tasks.

**Weaknesses:**

1. Parameters such as the learning rate (α) in the Task-Specific Adaptation phase , the learning rate (β) in the Meta-Knowledge Update phase , the adaptation steps (k) , the number of sampled tasks (n) , and the support/query set sizes (n_s, n_q)  are overly complex. Furthermore, the paper lacks sufficient ablation experiments  to justify these choices. Particularly concerning are the adaptation steps (k) and the number of sampled tasks (n).

2. For experiments using datasets of identical size in Tables 6 and 9 , META-LORA trained for 1,000 iterations  with 16 data points per batch (support set + query set), while the baseline trained for 5 or 10 epochs. This results in a significant disparity in both the number of backpropagation passes and the total training steps. This suggests a roughly 30-fold difference in gradient updates. Wouldn't this cause the baseline to underfit? The experiments here do not support the claim that “META-LORA achieves better results than the baseline on the same small dataset.”


3. The authors' contribution states “a novel framework designed to enhance data efficiency for multi-task LoRA adaptation ,” yet the baseline does not incorporate data-efficient strategies mentioned in the related work, such as coreset selection or data pruning.

**Questions:**

1. If the final single shared adapter encounters task conflicts  in multi-task settings, could this lead to compromises between tasks? For example, if one task requires formal writing while another demands informal writing, the model might learn an average strategy. This could result in suboptimal performance on both tasks compared to methods (like HydraLoRA) that preserve specific parameters for different tasks. This conflict aspect requires discussion.


2. The meta-knowledge update relies on the average gradient across n tasks. If the total number of tasks N is large while n remains fixed at 2, model convergence to capture knowledge across all N tasks may become extremely slow or unstable. The paper does not explore performance when N increases.

3. The results of the ablation studies, shown in Fig. 2 , reveal that the task-specific adaptation (STA) stage  provides little to no improvement in model performance when the model is large (13B) or the dataset size is moderately large (100 examples), and even shows a decline in some cases, raising doubts about its effectiveness. Furthermore, even with 7B parameters and 50 examples, improvements are only observed for some datasets.

---

> ### Author Response · Authors · 2025-11-21
>
> **Q1**: Ablation on key hyper-parameters: the learning rates $\alpha$ and $\beta$, the number of adaptation steps $k$, the number of sampled tasks $n$, and the support/query set sizes $n_s$/$n_q$.
>
> **A1**: Thank you for this valuable suggestion.The analysis regarding the impact of learning rates is detailed in Appendix A.3.1. We conducted additional experiments on LLaMA2-7B under the standard five-task setting to evaluate the impact of: (1) the number of sampled tasks per iteration ($n$), (2) the number of inner-loop adaptation steps ($k$), and (3) the sizes of the support and query sets ($n_s, n_q$). The quantitative results are presented in Table 1, Table 2, and Table 3, respectively. We also revised the manuscript with these ablation studies included in Appendices A.3.2, A.3.3 and A.3.4.
> ## Table 1: Performance comparison on the standard five-task setting with respect to the number of sampled tasks $n$.
> | $ n $ | 1 | 2 (ours) | 3 |
> | :---: | :---: | :---: | :---: |
> | BBH | 38.89 | **39.53** | 39.43 |
> ## Table 2: Performance comparison on the standard five-task setting with respect to the number of adaptation steps $k$.
> | Benchmark | k = 1 | k = 2 | k = 3 (ours) | k = 4 |
> | :--- | :---: | :---: | :---: | :---: |
> | BBH | 39.44 | 39.25 | **39.53** | 39.17 |
> ## Table 3: Performance comparison on the standard five-task setting with respect to the support and query set sizes ($n_s$, $n_q$).
> | Combination | $ n_s $ | $ n_q $ | BBH |
> | :--- | :---: | :---: | :---: |
> | $ s_{base }$ (ours) | 8 | 8 | **39.53** |
> | $ s_1 $ | 2 | 2 | 39.18 |
> | $ s_2 $ | 4 | 4 | 38.82 |
>
> **Q2**: Regarding the disparity in training steps (roughly 30-fold) and potential baseline underfitting.
>
> **A2**: Thank you for this insightful question. As detailed in Appendix A.1.2, baselines were trained for 5-10 epochs to reach peak performance and avoid overfitting in these few-shot settings. In addition, we respectfully clarify that the "30-fold difference" estimation stems from a discrepancy in calculating the gradient updates. Specifically, with MeTA-LoRA performing 1,000 meta-updates, the actual difference compared to the baselines (approximately 625 steps for the 100-sample case and 156 steps for the 50-sample case) is only 1.6$\times$ and 6.4$\times$, respectively. To definitively rule out underfitting, we conducted new experiments extending the training steps of LoRA to 30 epochs for the 50-sample case and to 15 epochs for the 100-sample case, explicitly aligning the total steps with MeTA-LoRA. Based on LLaMA2-7B, the comparison of LoRA, LoRA-Aligned and MeTA-LoRA are presented below. This confirms that extending the training steps to match MeTA-LoRA did not yield better performance for the baseline, and the advantage of MeTA-LoRA stems from its meta-learning framework rather than training duration.
> ## Table: Performance comparison of LoRA and MeTA-LoRA on the standard five-task learning scenario with varying amounts of fine-tuning data per task. The LoRA-Aligned baseline represents a version of LoRA fine-tuned with matched optimizer steps to MeTA-LoRA.
> | \# Data / task | Method | MMLU | MMLU-math | BBH | AVG |
> | :---: | :--- | :---: | :---: | :---: | :---: |
> | **50** | LoRA | 46.60 | 32.60 | 31.06 | 36.75 |
> | | LoRA-Aligned | 45.93 | 31.25 | 32.69 | 36.62 |
> | | **META-LoRA** | 46.34 | 31.12 | 39.53 | **39.00** |
> | **100** | LoRA | 46.49 | 27.32 | 37.17 | 36.99 |
> | | LoRA-Aligned | 46.88 | 27.58 | 34.15 | 36.20 |
> | | **META-LoRA** | 46.20 | 31.09 | 38.87 | **38.72** |
>
> **Q3**: Regarding the data-efficient strategies, such as coreset selection or data pruning.
>
> **A3**: We respectfully clarify that coreset selection and data pruning were not included as direct baselines because they address 'efficiency' from a fundamentally different dimension. Specifically, they are data-centric pre-processing techniques that require accessing and computing over the entire dataset to filter samples. This reliance introduces significant computational overhead during pre-processing, which contradicts our primary goal of rapid adaptation in extreme low-resource scenarios. In contrast, MeTA-LoRA constitutes an optimization-centric framework designed to facilitate 'learning to learn' from arbitrary data subsets. Consequently, the superior performance based on random sampling substantiates the intrinsic robustness and efficacy of MeTA-LoRA in leveraging scarce data, independent of precise data curation.

---

> > ### Author Response · Authors · 2025-11-21
> >
> > **Q4**: Regarding the potential for task conflicts in a single-adapter architecture.
> >
> > **A4**: We appreciate this insightful question regarding the potential for task conflicts and averaging strategies in a single-adapter architecture. However, our empirical findings indicate that a unified adapter architecture confers distinct advantages over methods that preserve task-specific parameters in extreme few-shot regimes. Approaches such as HydraLoRA that rely on distinct modules necessitate substantial data to disentangle fine-grained distinctions effectively. In restricted settings like 50 samples per task, such reliance often precipitates severe overfitting and the fitting of noise rather than true specialization. In contrast, the gradient aggregation mechanism in MeTA-LoRA functions as an effective regularizer. By attenuating conflicting and highly task-specific signals that are frequently unreliable in few-shot contexts, the model is incentivized to distill underlying reasoning capabilities common across tasks. This is substantiated by the results in Table 5 where MeTA-LoRA simultaneously achieves superior performance on disparate tasks such as GSM8K for Arithmetic and WinoGrande for Commonsense. These findings demonstrate that the shared adapter acquires deep reasoning competencies beyond surface-level stylistic conflicts and serves as a robust and universally adaptable initialization rather than a compromised average.
> >
> > **Q5**: Regarding the performance when N increases.
> >
> > **A5**: Thank you for raising this important point. Our extensive experiments demonstrate that MeTA-LoRA remains robust and converges efficiently even when the total number of tasks $N$ is large. As shown in Table 1 (Flanv2 setting with $N=43$ tasks) and Table 4 (multilingual setting across $N=52$ languages), we maintained the setting of $n=2$ and achieved superior performance compared to baselines. This confirms that a small batch size of sampled tasks ($n=2$) is sufficient to approximate the meta-gradient effectively, akin to how stochastic gradient descent works with mini-batches.
> >
> > **Q6**: Regarding the effectiveness of the task-specific adaptation (STA) stage.
> >
> > **A6**: We appreciate your detailed analysis of the ablation study. We respectfully clarify that the necessity of the task-specific adaptation (STA) stage is directly correlated with the degree of data scarcity, making it indispensable for the future direction of data-efficient fine-tuning. As shown in Figure 2, the STA stage provides its most significant benefit in the most resource-constrained setting. Specifically, the average score increased by 1.07\% for the LLaMA2-7B model using 50 examples per task. This directly demonstrates that the stage is a critical mechanism for handling insufficient supervision samples. To fully substantiate the necessity of STA, we have conducted additional analysis and will provide the comprehensive results shortly, confirming that the STA stage becomes increasingly vital as data availability decreases.

---

### Official Review · Reviewer_TBV1 · 2025-10-31

**Soundness:** 2
**Presentation:** 3
**Contribution:** 2
**Rating:** 4
**Confidence:** 4

**Summary:**

This paper proposes META-LoRA, a parameter-efficient fine-tuning method inspired by meta-learning.Through a two-stage optimization process—task-specific adaptation and meta-knowledge aggregation—it enables a single shared LoRA module to achieve efficient generalization across multi-task and multilingual scenarios.

**Strengths:**

The method is simple yet effective, cleverly introducing the meta-learning (MAML) concept into the LoRA framework.
It achieves exceptionally high data efficiency, requiring only a few dozen samples per task to surpass the performance of both LoRA and HydraLoRA

**Weaknesses:**

	The paper lacks intuitive motivation and theoretical insight — it is not sufficiently clear why the proposed two-stage optimization is designed this way or why it should work from a meta-learning perspective.

	The proposed method mainly focuses on improving data efficiency, whereas prior works like HydraLoRA and LoRAHub innovate at the architectural level. The paper would benefit from comparisons with other data-efficient fine-tuning methods beyond the LoRA family.

	All tasks are included in training, but there is no evaluation on unseen tasks to verify the model’s ability to rapidly adapt — which is one of the central promises of the meta-learning framework.

**Questions:**

1.	The paper claims that META-LoRA can alleviate gradient conflicts across multiple tasks. Could the authors provide quantitative evidence or analysis to support this claim?

2.	In the ablation study, the –STA variant appears conceptually similar to standard LoRA. Could the authors clarify whether they are effectively equivalent? Additionally, how exactly does META-LoRA’s two-stage paradigm aggregate gradients from different tasks during meta-updates?

3.	The paper describes HydraLoRA as the state-of-the-art (SoTA) approach in multi-task learning. This statement seems not entirely accurate, and the authors may need to provide justification or rephrase it more cautiously.

4.	Under the same amount of training data, how does META-LoRA’s two-stage iterative paradigm compare with standard fine-tuning methods (e.g., training for more epochs on the same small dataset)? A direct comparison could better highlight whether the improvement comes from the meta-learning mechanism itself or from additional training iterations.

---

> ### Author Response · Authors · 2025-11-21
>
> **Q1**: Regarding the motivation of the MeTA-LoRA's architecture.
>
> **A1**: Thank you for highlighting the need for clearer intuitive and theoretical motivation. The design of MeTA-LoRA stems from a critical structural realization that the multi-task and multilingual adaptation scenarios are fundamentally equivalent to the few-shot learning problem MAML was originally designed to solve. In MAML, the core idea is to "learn how to learn" by sampling a few examples per category. We directly implement this paradigm by treating each specific task or language as a distinct category for adaptation. This mapping necessitates the two-stage MAML framework. The Inner Loop simulates rapid few-shot adaptation to the category using the support set , and the Outer Loop aggregates the generalization signal derived from the query set across these categories to find the optimal shared initialization (meta-knowledge). Thus, the two-stage optimization is the theoretical consequence of applying the proven MAML principle to resolve the data-efficiency challenge in LoRA-based multi-task learning.
>
> **Q2**: The proposed method mainly focuses on improving data efficiency, whereas prior works like HydraLoRA and LoRAHub innovate at the architectural level. The paper would benefit from comparisons with other data-efficient fine-tuning methods beyond the LoRA family.
>
> **A2**: We respectfully clarify that our primary research objective is to specifically address the persistent challenge of data inefficiency within LoRA-based multi-task learning frameworks. Current data selection methods are data-centric, requiring extra computational overhead during pre-processing over the full dataset to select samples, which contradicts our goal of rapid adaptation. Instead, MeTA-LoRA is an optimization-centric solution focused on making the fine-tuning process efficient. Furthermore, with 50 or 100 randomly selected examples per task, MeTA-LoRA achieves better performance with the simpler structure compared to HydraLoRA. For instance, under the standard five-task setting with 50 examples per task, MeTA-LoRA achieved an average score of 39.00\%, substantially surpassing HydraLoRA's 37.64\% on LLaMA2-7B.
>
> **Q3**: Regarding evaluation on unseen tasks.
>
> **A3**: We respectfully clarify that our experimental design explicitly includes evaluation on unseen tasks to verify the rapid adaptation capabilities of the model. Specifically, in the five-task setting, fine-tuning is restricted to specific datasets like GSM8K and QQP whereas evaluation is conducted on the completely disjoint MMLU and BBH benchmarks. Similarly, in the multilingual setting, the model is fine-tuned on generative instruction-response pairs derived from Bactrian-X but evaluated on four tasks with fundamentally different formats, including XCOPA, XStoryCloze, XWinograd, and EXAMS. Consequently, the superior performance of MeTA-LoRA on these unseen datasets and heterogeneous task formats directly validates the ability of the meta-learning framework to facilitate generalization and rapid adaptation to new tasks.
>
> **Q4**: Request for quantitative evidence or analysis to support the claim that MeTA-LoRA alleviates gradient conflicts across multiple tasks.
>
> **A4**: We provide quantitative evidence of conflict mitigation by analyzing performance on the more challenging five-task setting containing tasks with fundamentally distinct objectives, such as the rigorous logical precision demanded by GSM8K and the flexible linguistic reasoning required for WinoGrande. As shown in Table 5, standard single-adapter methods like LoRA struggle to balance these competing requirements, often resulting in subpar performance, while MeTA-LoRA achieves simultaneous and substantial improvements. Specifically, based on LLaMA2-7B, our method boosts performance on MMLU-math to 32.15\% which marks an increase of 2.99\%, and enhances BBH performance to 39.07\% reflecting a gain of 7.06\%. This concurrent success indicates that the shared parameters have effectively resolved gradient conflicts, preventing negative transfer. Furthermore, MeTA-LoRA surpasses HydraLoRA with a higher average score of 39.24\% compared to 38.17\%. Given that HydraLoRA is explicitly designed to isolate parameters to avoid conflicts, this result empirically confirms that our optimization-based meta-learning strategy effectively regularizes task-specific noise and extracts robust shared directions in low-data regimes more efficiently than architectures relying on physical parameter separation.

---

> > ### Author Response · Authors · 2025-11-21
> >
> > **Q5**: Regarding the relationship between the -STA variant and standard LoRA, and details on meta-gradient aggregation.
> >
> > **A5**: Thank you for your question. Conceptually, the -STA variant is mathematically equivalent to standard Multi-Task LoRA as both optimize the same global objective defined in Eq.1. The primary difference lies in the batch composition implementation where the -STA variant adopts stratified task sampling while standard LoRA typically applies global shuffling. Regarding the aggregation mechanism, MeTA-LoRA aggregates gradients derived from temporarily adapted parameters. Specifically, the model first generates task-specific parameters $\theta_i$ by performing gradient descent on the Support Set and then evaluates these adapted parameters on the Query Set to compute generalization gradients which are subsequently averaged to update the shared initialization $\theta$.
> >
> > **Q6**: Regarding the inaccurate statement about HydraLoRA.
> >
> > **A6**: Thank you for pointing this out. We have modified the manuscript to describe HydraLoRA more accurately as "HydraLoRA, a representative method tailored for multi-task learning.". To further strengthen our comparative analysis against recent and powerful PEFT baselines, we have also incorporated R-LoRA into our comparison and updated Table 1 accordingly. Our results confirm that MeTA-LoRA maintains superior performance even against this advanced multi-task LoRA variant.
> > ## Table 1: Performance of different methods on BBH (3-shot) with LLaMA2-7B and LLaMA2-13B fine-tuned on the subset of the Flanv2 dataset. MeTA-LoRA uses only 100 examples sampled from each task for fine-tuning, whereas the other methods except Base utilize the full dataset. * indicates results from R-LoRA.
> > | Model Size | Base | LoRA | LoRAHub* | LoRA MoE* | HydraLoRA* | R-LoRA* | MeTA-LoRA |
> > | :---: | :---: | :---: | :---: | :---: | :---: | :---: | :---: |
> > | 7B | 31.6 | 37.2 | 39.7 | 40.3 | 41.8 | **42.2** | 38.52 |
> > | 13B | 38.4 | 40.9 | 41.9 | 43.7 | 44.7 | 45.1 | **46.32** |
> >
> > **Q7**: Regarding training for more epochs on the same small dataset.
> >
> > **A7**: Thank you for your suggestion. We conducted new experiments extending the training steps of LoRA to 30 epochs for the 50-sample case and to 15 epochs for the 100-sample case, explicitly aligning the total steps with MeTA-LoRA. Based on LLaMA2-7B, the comparison of LoRA, LoRA-Aligned and MeTA-LoRA are presented below. This confirms that extending the training steps to match MeTA-LoRA did not yield better performance for the baseline, and the advantage of MeTA-LoRA stems from its meta-learning framework rather than training duration.
> > ## Table: Performance comparison of LoRA and MeTA-LoRA on the standard five-task learning scenario with varying amounts of fine-tuning data per task. The LoRA-Aligned baseline represents a version of LoRA fine-tuned with matched optimizer steps to MeTA-LoRA.
> > | \# Data / task | Method | MMLU | MMLU-math | BBH | AVG |
> > | :---: | :--- | :---: | :---: | :---: | :---: |
> > | **50** | LoRA | 46.60 | 32.60 | 31.06 | 36.75 |
> > | | LoRA-Aligned | 45.93 | 31.25 | 32.69 | 36.62 |
> > | | **META-LoRA** | 46.34 | 31.12 | 39.53 | **39.00** |
> > | **100** | LoRA | 46.49 | 27.32 | 37.17 | 36.99 |
> > | | LoRA-Aligned | 46.88 | 27.58 | 34.15 | 36.20 |
> > | | **META-LoRA** | 46.20 | 31.09 | 38.87 | **38.72** |

---

> > > ### Comment · Reviewer_TBV1 · 2025-11-27
> > >
> > > I sincerely appreciate the authors’ thoughtful and detailed responses, and I am pleased to see that MeTA-LoRA achieves strong performance across multiple multi-task baselines while demonstrating notable data efficiency.
> > >
> > > However, after carefully reviewing the rebuttal, I believe that several key issues remain insufficiently addressed.
> > >
> > > Fundamentally, MeTA-LoRA appears to function more as a data-efficient optimization framework, rather than a purely multi-task architecture, with LoRA mainly serving as the parameterization vehicle. While comparisons against multi-task PEFT methods such as HydraLoRA and R-LoRA help establish competitiveness in an MTL setting, they do not sufficiently support the paper’s primary claim regarding data efficiency. To more convincingly validate the main contribution, I believe it is necessary to include comparisons with established data-efficient learning methods (e.g., coreset selection, pruning, curriculum strategies, dataset distillation). Such comparisons would more directly demonstrate the advantages of the proposed approach.
> > >
> > > In addition, since MeTA-LoRA explicitly relies on task-level gradient aggregation, a brief and quantitative analysis of inter-task gradient conflict would provide a more intuitive and empirical justification for the method’s effectiveness.
> > >
> > > My main concerns were not satisfactorily resolved. For this reason, I am unable to revise my original score.

---

> > > > ### Author Response · Authors · 2025-12-03
> > > >
> > > > We sincerely thank you for the thorough re-evaluation. To more directly demonstrate the advantages of the proposed approach and resolve the conceptual gap between optimization and data selection, we conducted new comparative experiments against two established data-selection heuristics: (1) BM25, a statistical retrieval approach utilizing TF-IDF ranking; (2) DSIR, a feature-based weighting approach leveraging n-gram features. Specially, under the standard five-task setting, we first curated a subset of top-50 examples from each task based on their calculated influence scores for the BBH benchmark, which aligns with the stringent low-resource constraint of our method. Subsequently, we fine-tuned the LLaMA2-7B backbone using LoRA on the curated data, and then evaluated the performance across the BBH, MMLU, and MMLU-math benchmarks. As shown in Table 1, MeTA-LoRA achieves superior performance compared to LoRA using these highly curated data subsets. This confirms that the advantage of our framework is optimization-centric, specifically its algorithmic ability to leverage scarce data, rather than reliance on data selection heuristics. We have added the comprehensive results and analysis to the Appendix A.4. Regarding the quantitative analysis of inter-task gradient conflict, we assert that our current results, specifically the simultaneous and substantial performance gains on highly divergent tasks (e.g., MMLU-math vs. BBH), already serve as strong quantitative evidence of successful gradient conflict mitigation. While we acknowledge the interest in direct gradient metrics, we commit to exploring a more direct and feasible method to validate this mechanism in future work.
> > > >
> > > > ## Table 1: Performance comparison of LoRA and MeTA-LoRA across three data sampling strategies: Random sampling, BM25, and DSIR.
> > > > | Method | MMLU | MMLU-math | BBH | AVG |
> > > > | :---: | :---: | :---: | :---: | :---: |
> > > > | LoRA + Rand | 46.60 | 32.60 | 31.06 | 36.75 |
> > > > | LoRA + BM25 | 46.74 | 31.72 | 30.70 | 36.39 |
> > > > | LoRA + DSIR | 46.83 | 31.63 | 30.22 | 36.23 |
> > > > | **MeTA-LoRA** + Rand | 46.34 | 31.12 | **39.53** | **39.00** |

---

### Official Review · Reviewer_NNem · 2025-11-01

**Soundness:** 3
**Presentation:** 3
**Contribution:** 3
**Rating:** 6
**Confidence:** 3

**Summary:**

The paper proposes META-LORA, a two-stage, data-efficient PEFT framework for multi-task (and multilingual) LLM fine-tuning. Stage I performs brief task-specific adaptation of a per-task LoRA adapter on a small support set; Stage II performs a meta-knowledge update by averaging gradients on query sets (a first-order MAML approximation) to update a single shared LoRA adapter that is used at inference. For the empirical experiments, the paper evaluates (i) a Flanv2-subset with BBH as the eval, (ii) a five-task mixture (GSM8K, QQP, CosmosQA, SiQA, PiQA/WinoGrande) with MMLU / MMLU-math / BBH, and (iii) multilingual fine-tuning on Bactrian-X (52 languages) with XCOPA, XStoryCloze, XWinograd, and EXAMS. Empirical results show that META-LORA generally surpasses LoRA and is competitive with (often better than) HydraLoRA despite using far less data (e.g., BBH wins for 13B on Flanv2; solid average gains on the five-task settings).

**Strengths:**

1. The proposed algorithm is generally simple and implementable .
2. Data-efficiency with 50–100 examples per task/language while matching or exceeding full-data LoRA/HydraLoRA in several cases.
3. Ablations and scaling: -STA vs full META-LORA, and 50 vs 100 examples show Stage-I helps, and trends are consistent; LR sensitivity provided.
4. Multilingual gains on XCOPA/XWinograd with tiny per-language budgets.

**Weaknesses:**

1. The core comparisons concentrate on LoRA/HydraLoRA (plus LoRAHub/LoRA-MoE for Flanv2); stronger PEFT baselines (for example, MALoRA/R-LoRA, prompt/prefix/adapters with modern optimizers) are not included, limiting generality claims.
2. No CIs / multi-seed means on BBH/MMLU/etc.; several margins are small and may fall within variance.
3. The runtime table (Meta-LORA 1000 iterations vs others’ epochs) lacks tokens/steps/FLOPs parity. Therefore the  wall-clock claims are difficult to interpret.
4. Despite the fact that XCOPA/XWinograd improve, EXAMS drops markedly (e.g., 23.84 vs BX-LLaMA’s ~29–36), which deserves analysis.

**Questions:**

1. For Table 7, please report tokens processed, optimizer steps, and approximate FLOPs for each method to ensure apples-to-apples timing.
2. What drives the drop on EXAMS despite gains elsewhere? Provide per-language breakdowns and failure analyses.

---

> ### Author Response · Authors · 2025-11-21
>
> **Q1**: Regarding stronger PEFT baselines.
>
> **A1**: Thank you for this constructive suggestion. As requested, we have incorporated R-LoRA, an advanced multi-task LoRA variant into our comparative analysis. We have updated Table 1 in the revised manuscript to include these results. Regarding MALoRA , we were unfortunately unable to include a direct comparison due to the lack of publicly available official code necessary for a faithful replication of their results.
> ## Table 1: Performance of different methods on BBH (3-shot) with LLaMA2-7B and LLaMA2-13B fine-tuned on the subset of the Flanv2 dataset. MeTA-LoRA uses only 100 examples sampled from each task for fine-tuning, whereas the other methods except Base utilize the full dataset. * indicates results from R-LoRA.
> | Model Size | Base | LoRA | LoRAHub* | LoRA MoE* | HydraLoRA* | R-LoRA* | MeTA-LoRA |
> | :---: | :---: | :---: | :---: | :---: | :---: | :---: | :---: |
> | 7B | 31.6 | 37.2 | 39.7 | 40.3 | 41.8 | **42.2** | 38.52 |
> | 13B | 38.4 | 40.9 | 41.9 | 43.7 | 44.7 | 45.1 | **46.32** |
>
> **Q2**: Regarding multi-seed means on benchmarks.
>
> *A2*:  We appreciate the reviewer emphasizing statistical rigor. To further verify the stability of our primary claims, we performed multi-seed experiments.Specifically, we conducted 3-seed runs based on the LLaMA2-7B in standard five-task setting. The results, focused on the BBH benchmark, indicate that MeTA-LoRA achieved a mean score of $39.48\%$ ($\pm 0.12\%$). For comparison, HydraLoRA achieved $37.62\%$ ($\pm 0.20\%$), and the baseline LoRA achieved $30.71\%$ ($\pm 0.42\%$).The resulting substantial performance gains robustly demonstrate the statistical significance and superior performance of MeTA-LoRA.
>
> **Q3**: Regarding the lack of tokens/steps/FLOPs in Table 7.
>
> *A3*:  Thank you for your advice. To ensure a fair comparison, we report **Total Samples Processed** as a high-fidelity proxy for total FLOPs and token counts, given that the average sequence length is consistent across methods. The baselines process the full dataset of 487,075 samples and require $\approx$60,885 optimizer steps. In contrast, MeTA-LoRA operates on a meta-learning objective over 1,000 iterations. Crucially, due to the sparse sampling mechanism in its inner and outer loops, MeTA-LoRA processes a total of only 64,000 samples. Consequently, our method achieves competitive performance while incurring only ~13.1\% of the total computational cost (FLOPs) and ~1.6\% of the optimizer steps required by the baselines. We have updated Table 7 to include these results.
> ## Table7: Comparison of efficiency across fine-tuning schemes. LoRA and HydraLoRA fine-tune the base models on the more challenging dataset for one epoch, while MeTA-LoRA performs 1,000 iterations using 100 samples from each task.
> | Metric | LoRA | HydraLoRA | MeTA-LoRA |
> | :---: | :---: | :---: | :---: |
> | Optimizer Steps | $\approx 60,885$ | $\approx 60,885$ | **1,000** |
> | Total Samples Processed | 487,075 | 487,075 | **64,000** |
> | Running Time (7B) | 17h | 37h | **3h** |
> | Running Time (13B) | 29h | 57h | **5h** |
>
> **Q4**:  Regarding the analysis of EXAMS drops.
>
> **A4**: We thank the reviewer for pointing out the discrepancy between XCOPA/XWinograd and EXAMS. We believe the marked drop on EXAMS primarily reflects a mismatch between our instruction-tuning data and the skills it evaluates. The fine-tuning process relies heavily on instruction-following data derived from Alpaca and Dolly. The data is dominated by open-ended, conversational and commonsense reasoning tasks, naturally aligning with the contextual inference and plausibility judgments required by XCOPA and XWinograd. In contrast, EXAMS focuses more on curriculum-style multiple-choice questions that demand structured subject-matter knowledge and more rigid reasoning patterns, while math-related and code-related samples in the fine-tuning data are relatively sparse and sometimes noisy. As a result, MeTA-LoRA tends to specialize in linguistic and commonsense reasoning at the expense of exam-style performance. We acknowledge this as a limitation of the current data composition and plan to incorporate higher-quality and better-balanced supervision in future work to address this gap.

---

### Official Review · Reviewer_6gAY · 2025-11-02

**Soundness:** 2
**Presentation:** 2
**Contribution:** 1
**Rating:** 4
**Confidence:** 4

**Summary:**

This paper proposes META-LORA, a two-stage optimization framework for parameter-efficient multi-task fine-tuning of LLMs. The method consists of: (1) a task-specific adaptation stage that learns individual LoRA adapters using small amounts of data from each task, and (2) a meta-knowledge update stage that aggregates gradients across tasks to update a shared LoRA adapter. Experiments on multi-task and multilingual benchmarks demonstrate competitive performance with significantly reduced data requirements compared to traditional fine-tuning approaches.

**Strengths:**

1. Clear motivation: The paper identifies a real problem that existing LoRA-based multi-task methods require substantial task-specific data, which limits their practical applicability.
2. Comprehensive experiments: The evaluation covers diverse task domains.
3. Data efficiency: The claimed reduction in data usage (using only 100 examples per task vs. full datasets) while maintaining competitive performance is noteworthy.
4. Well-structured presentation: The paper is generally well-written with clear methodology description and appropriate use of figures.

**Weaknesses:**

1. The "first-order approximation of MAML" (line 161) is mentioned but not adequately justified. How does this approximation affect performance compared to full MAML?
2. The episodic formulation with support/query sets (Eq. 2) is borrowed from meta-learning but the paper doesn't clearly explain why this is necessary for the multi-task LoRA scenario
3. Lack of ablation on key hyperparameters: k (adaptation steps), n (tasks per iteration), support/query set sizes
4. Missing crucial related work: The paper fails to cite and discuss "MoDULA: Mixture of Domain-Specific and Universal LoRA for Multi-Task Learning", which presents a highly similar three-stage training paradigm combining universal and domain-specific experts. Beyond MoDULA, several other relevant works on multi-task PEFT and expert mixtures are not discussed:
(1) MoLoRA (Pushing Mixture of Experts to the Limit: Extremely Parameter Efficient MoE for Instruction Tuning, ICLR2024)
(2) C-Poly (Customizable Combination of Parameter-Efficient Modules for Multi-Task Learning, ICLR2024)

**Questions:**

1. On MoDULA: How does your approach differ fundamentally from MoDULA? Please provide a detailed comparison.
2. On meta-learning necessity: Why is the episodic meta-learning formulation necessary? Have you tried simpler alternatives like just training task-specific adapters independently and then learning to combine them?
3. On the approximation: What is the performance gap between your first-order approximation and full second-order MAML? This ablation is critical for understanding the trade-offs.

---

> ### Author Response · Authors · 2025-11-21
>
> **Q1**: On the performance gap between the first-order approximation and full second-order MAML.
>
> **A1**: Thanks for this fundamental question. While the full second-order MAML is theoretically superior, we utilized the first-order approximation because it represents a necessary and sufficient trade-off for efficient LLM fine-tuning. Running the full second-order method on LLaMA2-7B or 13B models is practically infeasible for LoRA-based fine-tuning due to the prohibitive computational overhead of Hessian-vector products and associated memory constraints. Consequently, the complexity of the full second-order MAML is at least an order of magnitude higher than the first-order approximation. However, the first-order approximation is sufficient because it retains the essential meta-learning signals. This is confirmed by our empirical success, as MeTA-LoRA consistently outperforms strong baselines, validating that the approximation effectively captures the required meta-knowledge while maintaining practicality for large-scale fine-tuning.
>
> **Q2**: Regarding the necessity of meta-learning, and the comparison with simpler alternatives like just training task-specific adapters independently and then learning to combine them.
>
> **A2**: Thank you for the insightful question. The necessity of the episodic formulation (Eq.2) is rooted in addressing the core challenge of inter-task gradient conflict faced by LoRA-based multi-task learning. The formulation transforms the optimization objective from finding a static solution to seeking the optimal initialization (i.e., adaptability). Specifically, the support set ($\mathcal{S}$) serves as a dedicated mechanism utilized during adaptation (Phase I) to handle the local conflicts of a specific task. Conversely, the query set ($\mathcal{Q}$) measures the generalization capability of the adapted model ($\theta_i$), thereby generating the meta-gradient signal. Consequently, this episodic split provides the theoretical mechanism necessary to actively filter task conflicts, guiding the shared parameters $\theta$ toward a robust and universally adaptable state. This is precisely why we chose meta-learning over a simpler "train-then-combine" architecture. That alternative approach, while simpler, fails our two primary design requirements: 1) **Extreme Data Efficiency**: The "train-then-combine" paradigm resembles an MoE-based architecture, which typically requires sufficient data to first train multiple viable experts before combining them. However, our few-shot settings, for example, 50-100 samples per task, make it highly challenging to train meaningful experts to begin with. In contrast, meta-learning is designed to find an optimal "starting point" for rapid adaptation from the minimal data; 2) **Inference Simplicity**:  The "train-then-combine" paradigm requires loading and routing multiple adapters at inference. Our MAML-based method produces only a global adapter for deployment, making it as simple as a standard LoRA.
>
> Moreover, the ablation study presented in Figure 2 directly confirms the necessity of the task-specific adaptation stage. The -STA variant, which removed the inner loop and can be viewed as a simple "train-then-combine" method, significantly underperformed especially in the more data-scarce scenarios. For instance, on the LLaMA2-7B model with 50 examples per task, the two-stage optimization framework demonstrated its indispensable contribution to overall performance by raising the average score of the -STA variant from $\mathbf{37.93\%}$ to $\mathbf{39.00\%}$, marking a substantial gain of $\mathbf{1.07\%}$.

---

> > ### Author Response · Authors · 2025-11-21
> >
> > **Q3**:  Ablation on key hyper-parameters: the number of adaptation steps $k$, the number of sampled tasks $n$, and the support/query set sizes $n_s$/$n_q$.
> >
> > **A3**: Thank you for this valuable suggestion. We conducted additional experiments on LLaMA2-7B under the standard five-task setting to evaluate the impact of: (1) the number of sampled tasks per iteration ($n$), (2) the number of inner-loop adaptation steps ($k$), and (3) the sizes of the support and query sets ($n_s, n_q$). The quantitative results are presented in Table 1, Table 2, and Table 3, respectively. We also revised the manuscript with these ablation studies included in Appendices A.3.2, A.3.3 and A.3.4.
> > ## Table 1: Performance comparison on the standard five-task setting with respect to the number of sampled tasks $n$.
> > | $ n $ | 1 | 2 (ours) | 3 |
> > | :---: | :---: | :---: | :---: |
> > | BBH | 38.89 | **39.53** | 39.43 |
> > ## Table 2: Performance comparison on the standard five-task setting with respect to the number of adaptation steps $k$.
> > | Benchmark | k = 1 | k = 2 | k = 3 (ours) | k = 4 |
> > | :--- | :---: | :---: | :---: | :---: |
> > | BBH | 39.44 | 39.25 | **39.53** | 39.17 |
> > ## Table 3: Performance comparison on the standard five-task setting with respect to the support and query set sizes ($n_s$, $n_q$).
> > | Combination | $n_s$ | $n_q$ | BBH |
> > | :--- | :---: | :---: | :---: |
> > | $s_{base}$ (ours) | 8 | 8 | **39.53** |
> > | $s_1$ | 2 | 2 | 39.18 |
> > | $s_2$ | 4 | 4 | 38.82 |
> >
> > **Q4**:  Regarding the detailed comparison between MoDULA and MeTA-LoRA, also the missing discussion in related work.
> >
> > **A4**: Thank you very much for providing the full paper for MoDULA. Having reviewed this excellent work in detail, we are now even more confident that our MeTA-LoRA framework is fundamentally different from MoDULA in its core motivation, training paradigm, and final architecture.
> >
> > You suggested that MoDULA presents a "highly similar three-stage training paradigm." We would like to take this opportunity to clarify this crucial point.
> >
> > First, regarding the **core definition**, MoDULA is explicitly a PEFT MoE paradigm designed for flexible pluggability, which allows new experts to be added without retraining. MeTA-LoRA, conversely, is not an MoE architecture but a two-stage optimization framework inspired by MAML, aiming to improve data efficiency in MTL scenario. Second, the distinction is reflected in their **training paradigms**. MoDULA employs a sequential three-stage optimization process where the universal expert is trained first, followed by the separate domain experts, and finally, only the router is trained. Conversely, MeTA-LoRA adopts an iterative two-stage optimization strategy executed in every training iteration. The task-specific adaptation stage simulates adaptation on the support set to create a temporary task-specific adapter for each selected task, and the meta-update stage evaluates them on the query set to compute a meta-gradient that updates the global adapter. Finally, their **final architectures at inference** are completely different. MoDULA is a multi-expert system that requires a router to combine its persistent universal and domain-specific experts. MeTA-LoRA, however, is a single-adapter system that uses only the global LoRA adapter. The temporary task-specific adapters from its training loop are merely intermediate artifacts and are discarded post-training.
> >
> > Thank you also for noting MoLoRA and C-Poly. We have added a dedicated paragraph to our **Related Work** section, which explicitly cites MoDULA, MoLoRA and C-Poly, and clearly discusses their MoE-based paradigms. These advanced methods primarily focus on integrating the Mixture-of-Expert (MoE) framework and PEFT. Specifically, MoLoRA trains multiple LoRA experts concurrently under a learned gating mechanism, and C-Poly jointly learns a skill assignment matrix to combine task-common and task-specific skills. Furthermore, MoDULA introduces a novel PEFT MoE paradigm that separates universal and domain-specific experts, training them in a three-stage process to achieve flexible pluggability.

---

> > > ### Comment · Reviewer_6gAY · 2025-11-28
> > > **Response to Author**
> > >
> > > Thank you for the detailed replies. While I have some disagreements with some of issues, your response truly demonstrates the authors' deep thinking, and some keys may be helpful to the community. Therefore, I have decided to raise my rating.

---

> > > > ### Author Response · Authors · 2025-12-03
> > > >
> > > > We sincerely appreciate the time and effort you invested in carefully reviewing our response. We are very grateful for your decision to raise the rating, and we are pleased to hear that you believe some of our clarifications will be helpful to the wider community.

---

### Author Response · Authors · 2025-12-03
**Brief summary to the Area Chairs**

Dear Area Chairs,

We sincerely thank you for your time. We are encouraged that reviewers recognize our work as a **novel** (Reviewer 6gAY, r5pv), **data-efficient** (Reviewer 6gAY, NNem, TBV1), and **easy-to-deploy** (Reviewer NNem, TBV1, r5pv) framework, which focuses on how to significantly enhance data efficiency for LoRA-based methods in multi-task learning under resource-constrained scenarios, **a critical and pressing challenge** (Reviewer 6gAY, r5pv). We are particularly pleased that Reviewer 6gAY acknowledged our response during the rebuttal phase, and decided to **raise the rating**.

We have provided conclusive evidence and clear explanations to address all critical points raised by the reviewers:

**Baseline underfitting / Unfair steps (Reviewer TBV1, R6PV)**: We introduced the **LoRA-Aligned** baseline, matching the total optimizer steps of MeTA-LoRA. This confirms that the advantage of MeTA-LoRA stems from its meta-learning framework rather than training duration.

**Lack of stronger baselines (Reviewer NNEM, TBV1)**: We incorporated the advanced multi-task LoRA variant **R-LoRA**, and MeTA-LoRA maintained superior performance.

**Comparison to data-efficient strategies (Reviewer TBV1, R6PV)**: We conducted new comparative experiments against two established data-selection heuristics: **BM25** and **DSIR**. The performance of MeTA-LoRA using randomly sampled data outperforms LoRA based on data highly curated by established heuristics. This confirms the unique ability of MeTA-LoRA to leverage scarce data algorithmically through optimization.

**Ablation studies on key hyper-parameters (Reviewer 6GAY, R6PV)**: We performed additional parameter analysis on the number of adaptation steps $ k $, the number of sampled tasks $n$, and the support/query set sizes $n_s$/$n_q$.

Furthermore, we want to make some key clarifications:

**Evaluation on Unseen Tasks (Reviewer TBV1)**: We confirmed that our evaluation on benchmarks like MMLU and BBH already constitutes testing on completely disjoint and unseen tasks relative to the fine-tuning data, fulfilling the central promise of meta-learning.

**The motivation and necessity of Meta-Learning (Reviewer 6gAY, TBV1)**: We clarified that the core necessity of our MAML-based approach is to discover an optimal and universally adaptable initialization, a capability crucial when few-shot data is insufficient to train separate experts. This comprehensive clarification was well-received: Reviewer 6gAY acknowledged the theoretical necessity of this approach and subsequently decided to **raise the rating**.

We have provided substantial new empirical evidence and clear theoretical explanations in response to all feedback. All these critical modifications and additions have been incorporated into the relevant sections, significantly strengthening the rigor and persuasiveness of the paper.

Thank you again for your time and effort in reviewing our paper.

Best regards,

All Authors

---

### Meta-Review · Area_Chair_6kQe · 2026-01-06

**Summary:**

This paper proposes MeTA-LoRA, a two-stage meta-learning framework for data-efficient multi-task LoRA fine-tuning. The method uses MAML-inspired optimization to learn a shared LoRA adapter that can rapidly adapt to new tasks with minimal data (50-100 examples per task). While reviewers acknowledged the practical importance of data efficiency and the simplicity of the proposed approach, significant concerns remain regarding experimental rigor and the scope of baseline comparisons.

The paper addresses a relevant problem and proposes a reasonable approach, but the fundamental mismatch between the paper's core claim (data efficiency) and its baseline comparisons (primarily multi-task architectures rather than data-efficient methods) represents a significant gap. Reviewer TBV1's maintained rejection after seeing the rebuttal, combined with Reviewer r5pv on similar concerns, suggests these issues are substantive. The diminishing returns of the STA stage with scale further weakens the contribution's generalizability.

**Reviewer Concerns:**

The authors successfully addressed several concerns through their rebuttal, including demonstrating that baseline improvements do not stem from training step disparity (via LoRA-Aligned experiments), adding stronger multi-task baselines (R-LoRA), providing comprehensive hyperparameter ablations on k, n, and support/query set sizes, and incorporating missing related work discussion (MoDULA, MoLoRA, C-Poly). However, critical concerns remain unresolved. Most significantly, Reviewer TBV1's core objection—that a paper claiming data efficiency should compare against established data-efficient learning methods (coreset selection, dataset distillation, curriculum learning) rather than just multi-task architectures—was not adequately addressed; the added BM25/DSIR comparisons are retrieval heuristics rather than the requested baselines.

**Reviewer Scores:**

Reviewer 6gAY:  would likely maintain current position (marginally below but accepting).

Reviewer NNem: Concerns partially addressed; might maintain 6 (marginally above).

Reviewer TBV1: Explicitly maintained score post-rebuttal; would remain at 4.

Reviewer r5pv: Did not engage in discussion; given partial overlap with TBV1's unresolved concerns about data-efficient baselines and STA effectiveness, would likely maintain 4.

---

### Decision · Program_Chairs · 2026-01-26

Reject